# Sulfane Sulfur Regulates LasR-Mediated Quorum Sensing and Virulence in *Pseudomonas aeruginosa* PAO1

**DOI:** 10.3390/antiox10091498

**Published:** 2021-09-21

**Authors:** Guanhua Xuan, Chuanjuan Lv, Huangwei Xu, Kai Li, Huaiwei Liu, Yongzhen Xia, Luying Xun

**Affiliations:** 1State Key Laboratory of Microbial Technology, Shandong University, 72 Binhai Road, Qingdao 266237, China; xuanguanhua@ouc.edu.cn (G.X.); chuanjuanlv@sdu.edu.cn (C.L.); huangweixu@mail.sdu.edu.cn (H.X.); likai03721@sdu.edu.cn (K.L.); liuhuaiwei@sdu.edu.cn (H.L.); 2School of Molecular Biosciences, Washington State University, Pullman, WA 99164-7520, USA

**Keywords:** quorum sensing, LasR, sulfane sulfur, signaling, protein persulfidation, virulence

## Abstract

Sulfane sulfur, such as inorganic and organic polysulfide (HS_n_^−^ and RS_n_^−^, *n* > 2), is a common cellular component, produced either from hydrogen sulfide oxidation or cysteine metabolism. In *Pseudomonas aeruginosa* PAO1, LasR is a quorum sensing master regulator. After binding its autoinducer, LasR binds to its target DNA to activate the transcription of a suite of genes, including virulence factors. Herein, we report that the production of hydrogen sulfide and sulfane sulfur were positively correlated in *P. aeruginosa* PAO1, and sulfane sulfur was able to modify LasR, which generated Cys^188^ persulfide and trisulfide and produced a pentasulfur link between Cys^201^ and Cys^203^. The modifications did not affect LasR binding to its target DNA site, but made it several-fold more effective than unmodified LasR in activating transcription in both in vitro and in vivo assays. On the contrary, H_2_O_2_ inactivates LasR via producing a disulfide bond between Cys^201^ and Cys^203^. *P. aeruginosa* PAO1 had a high cellular sulfane sulfur and high LasR activity in the mid log phase and early stationary phase, but a low sulfane sulfur and low LasR activity in the declination phase. Thus, sulfane sulfur is a new signaling factor in the bacterium, adding another level of control over LasR-mediated quorum sensing and turning down the activity in old cells.

## 1. Introduction

Hydrogen sulfide (H_2_S) has been proposed as a new gaseous signaling molecule, mediating various biological functions in mammals, including humans [1,2,3,4,5]. Owing to its signaling role in eukaryotes, H_2_S was suggested as a “clandestine microbial messenger” in 2006 [6]. Since then, several bacterial transcription factors, including FisR [7], SqrR [8], CstR [9], and CsoR [10], have been identified that indirectly respond to H_2_S, activating sulfur-oxidizing genes. First, sulfide:quinone oxidoreductase (SQR) converts H_2_S to sulfane sulfur, including inorganic and organic polysulfide (HS_n_^−^ and RS_n_^−^, *n* ≥ 2) and elemental sulfur (S^0^) [11,12]. Sulfane sulfur is then sensed by these transcription factors to turn on the transcription of sulfur-oxidizing genes, including *pdo* coding for persulfide dioxygenase (PDO). A synthetic gene circuit that combines SQR and CstR allows the host *Escherichia coli* to oxidize self-produced H_2_S to sulfane sulfur and then sense the latter, resulting in gene regulation in a manner similar to quorum sensing (QS) [13]. These examples highlight how H_2_S is converted by SQR to sulfane sulfur that is then sensed by these gene regulators.

Other gene regulators that are not involved in sulfur metabolism may also sense sulfane sulfur. OxyR, the H_2_O_2_-response gene regulator, senses cellular sulfane sulfur to turn on many genes, including those coding for the removal of excessive sulfane sulfur in *E. coli* [14]. MexR senses sulfane sulfur, which is maximally accumulated in late log phase by *Pseudomonas aeruginosa* PAO1, and activates the expression of a multidrug efflux pump for antibiotic resistance [15]. Additional examples are needed to confirm that H_2_S and sulfane sulfur are common signaling molecules in bacteria, regulating diverse microbial behaviors.

Heterotrophic bacteria routinely produce H_2_S and sulfane sulfur during normal growth [16,17]. L-Cysteine desulfhydrase directly converts L-cysteine to H_2_S and pyruvate [18]. Other enzymes, such as cystathionine γ-lyase (CSE), cystathionine β-synthase (CBS), cysteinyl-tRNA synthetase, and 3-mercaptopyruvate sulfurtransferase (3-MST), metabolize L-cysteine and its derivatives to sulfane sulfur, which can be further reduced to H_2_S [18,19,20,21]. H_2_S can also be generated through the reduction of sulfite by sulfite reductase (CysI) during sulfite and sulfate assimilation [22]. For bacteria with sulfide:quinone oxidoreductases (SQR), such as *P. aeruginosa* PAO1, self-produced sulfide is oxidized back to sulfane sulfur [16]. Sulfane sulfur is a regular cellular component in the plasma and cells of mammals, as well as inside bacteria [17,23]. In bacteria, the concentration of sulfane sulfur can be higher than 100 μM [17].

*Pseudomonas aeruginosa* is a ubiquitous Gram-negative bacterium and an opportunistic human pathogen [24]. *P. aeruginosa* strains are clinically significant, as several isolates are multidrug-resistant [25]. Various efforts, including using lavender essential oils, have been made to treat multidrug-resistant *P. aeruginosa* strains [26]. *P. aeruginosa* PAO1 has three QS systems, *las*, *rhl*, and *pqs* [27]. The QS systems regulate the expression of virulence factors, biofilm development, and production of secondary metabolites [28]. In the *las* system, LasI synthesizes the signal molecule N-(3-oxododecanoyl)-L-homoserine lactone (3O-C_12_-HSL). When LasR binds 3O-C_12_-HSL, it may bind to its targets, functioning as a transcription activator [29]. The *las* system regulates the *rhl* system that activates rhamnolipid biosynthesis and the *pqs* system, and the latter positively regulates pyocyanin biosynthesis [30,31,32,33]. Thus, LasR is a QS master regulator in *P. aeruginosa* PAO1.

Several factors affect LasR activity. The activity requires a threshold level of 3O-C_12_-HSL [34]. However, the saturating level of 3O-C_12_-HSL is not sufficient to fully induce the LasR regulon at low cell density [35,36,37]. Several proteins, e.g., QteE, QslA, and QscR, act to dampen gene activation by LasR [38], and some well-characterized LasR promoters use RpoS, a sigma factor for gene expression during the stationary phase [39]. The LasR activity is sensitive to oxidative stress [40]. Furthermore, surface association promotes the production of the small RNA Lrs1 that stimulates the production of LasR at low cell density [41]. Herein, we show that sulfide (H_2_S, HS^–^, and S_2_^–^) and sulfane sulfur also participate in LasR regulation in *P. aeruginosa* PAO1.

In this study, we deleted several genes involved in H_2_S production and oxidation in *P. aeruginosa* PAO1. The H_2_S-oxidizing mutant did not show any apparent differences in growth from the wild type, but the H_2_S-producing mutant displayed a clear reduction of several virulence factors that are activated by LasR. RNA-seq also indicated that LasR was less active in the H_2_S-producing mutant than in the wild type. Further analysis showed that LasR activity was significantly enhanced by sulfane sulfur, which was high in the mid log phase to stationary phase of growth. In the declination phase, cellular sulfane sulfur level became low, which was associated with significantly low LasR activity.

## 2. Materials and Methods

### 2.1. Strains, Plasmids, and Reagents

The strains and plasmids used in this work are listed in Appendix A. Unless noted otherwise, *P. aeruginosa* PAO1 and its derivatives were grown in lysogeny broth (LB) medium or Pseudomonas broth (PB) (2% Bacto-peptone, 0.14% MgCl_2_, 1% K_2_SO_4_, 2% glycerol) medium at 37 °C [42]. Kanamycin (50 μg/mL), ampicillin (100 μg/mL), or gentamicin (30 μg/mL) was added when required. Other chemicals such as NaHS (H_2_S donor) and 3O-C_12_-HSL were purchased from Sigma-Aldrich. HS_n_^−^ was prepared following a reported method [43], and the stock concentration was determined using a cyanolysis method and calibrated by using thiosulfate as the standard [43].

### 2.2. Gene Knockout and Complementation

The primers used for Pa*cbs*, Pa*cse*, Pa*mst*, Pa*cysI*, Pa*sqr1*, Pa*sqr2*, Pa*pdo,* and Pa*lasR* inactivation are listed in Appendix A. The deletions in *P. aeruginosa* PAO1 were performed according to a published method [44,45]. Briefly, about 1000-bp fragments upstream and downstream of the target gene were amplified from the PAO1 genomic DNA via PCR, linked, and cloned into pK18mobsacB_tet_ at the EcoRI site. The resulting plasmid was first transformed into *E. coli* and then transferred into *P. aeruginosa* PAO1 via conjugation. The integration into the chromosome of *P. aeruginosa* PAO1 by a homologous crossover was selected on agar plates of a chemically defined medium, with sodium gluconate as the sole carbon source containing tetracycline. The selection of the double crossover with 12% sucrose led to gene-deletion. For multiple deletions, the process was repeated. The deletion mutants, including Pa∆*cbs*∆*cse*∆*mst*∆*cysI*∆*sqr1*∆*sqr2*∆*pdo* (Pa7K), Pa∆*cbs*∆*cse*∆*mst*∆*cysI* (Pa∆H_2_S), Pa∆*sqr1*∆*sqr2*∆*pdo* (Pa3K), and Pa∆*lasR,* were confirmed by using colony PCR and DNA sequencing. For complementation, the target genes were amplified by PCR and cloned into linearized pBBR1MCS5 by using an In-Fusion HD cloning kit (Clontech, Mountain View, CA, USA). The resulting plasmids were then transferred into the PAO1 strain via electroporation.

### 2.3. Detection of H_2_S and Sulfane Sulfur

The production of H_2_S by PAO1 and its related mutant strains was detected with a paper strip with lead-acetate [Pb(Ac)_2_] (Shanghai, China) and a monobromobimane (mBBr) method [46]. Briefly, single colonies were innoculated in 2 mL of LB medium in a 15-mL glass tube, and a paper strip with lead-acetate was affixed at the top of the tube with a rubber stopper. The paper strip was examined and photographed to detect any lead(II)-sulfide black precipitates, indicting the production of H_2_S. H_2_S in the liquid culture was detected using the mBBr method [46]. Briefly, 5 μL of 25 mM mBBr was reacted with 50 μL of a sample at room temperature for 30 min in the dark, and an equal volume of 10% acetic acid in acetonitrile was added. The samples were centrifuged at 12,500× *g* for 2 min and the supernatant was analyzed using HPLC (LC-10AT, Shimadzu) with a fluorescence detector, as reported [11].

SSP4 (sulfane sulfur probe 4, 3′,6′-di(O-thiosalicyl)fluorecein) that reacts with sulfane sulfur to become fluorescent was used to check the relative levels of cellular sulfane sulfur in *P. aeruginosa* cells [47]. The cells were collected, washed with phosphate buffer saline, and resuspended in phosphate buffer saline at an OD_600nm_ of 1. Then 10 μM SSP4 and 0.5 mM hexadecyltrimethylammonium bromide (CTAB) were added to the sample, which was incubated in the dark at 30 °C, with shaking for 20 min. Cells were harvested by centrifugation and washed twice with phosphate buffer saline. The fluorescence of the resuspended cells (OD_600nm_ = 1) was detected by using a Synergy H1 microplate reader with excitation of 482 nm and emission of 515 nm. We were unable to quantify sulfane sulfur with SSP4. For quantification, cellular sulfane sulfur in *P. aeruginosa* PAO1 at different growth stages was reacted with sulfite to produce thiosulfate, which was then quantified according to a reported method [17]. Briefly, samples were mixed with the reaction buffer with sulfite to convert sulfane sulfur to thiosulfate by incubating at 95 °C for 20 min; buffer without sulfite was used as the control. The produced thiosulfate was detected by using the mBBr method [46], as briefly described above.

### 2.4. Rhamnolipids Production Measurement

Rhamnolipid production was measured by following a reported method [48]. Briefly, 5 μL of an overnight culture was placed onto a M8 minimal agar plate supplemented with 0.0005% (*m/v*) methylene blue and 0.02% (*m/v*) CTAB; the plate was incubated at 37 °C for 48 h before checking for the clear zone around a colony. For the H_2_S supplemental experiment, 10 drops of 1 mM NaHS in H_2_O were separately dropped on the plate cover. The plates were inverted, and H_2_S entered the medium via evaporation.

### 2.5. Pyocyanin Quantitation Assay

Pyocyanin concentration was determined by using a reported method [42]. Briefly, strains were grown at 37 °C in 5 mL of PB medium to stationary phase. The culture supernatants were extracted with 3 mL of chloroform, which was mixed with 1 mL of 0.2 M HCl to develop a pink to deep red color in the organic phase for measurements at 520 nm. The concentration was calculated and normalized to the cell density (OD_600nm_).

### 2.6. Lettuce Leaf Model of Infection

A lettuce leaf virulence assay was performed, as described [48,49]. Briefly, *P. aeruginosa* strains were grown in PB medium at 37 °C overnight with shaking (200 rpm); the cells were harvested, washed, and resuspended in sterile 10 mM MgSO_4_ to a bacterial density of 1 × 10^9^ cells/mL. Romaine lettuce leaves were washed with sterile distilled H_2_O with 0.1% bleach and then inoculated with the strains on the midribs of the leaves, which were placed in containers underlined with wet filter paper containing 10 mM MgSO_4_. The inoculated leaves were kept in a growth chamber at 37 °C. Symptoms were monitored daily. As a control, lettuce leaves were inoculated with 10 mM MgSO_4_.

### 2.7. Transcriptomic Analysis of PAO1 and PaΔH_2_S

For transcriptome sequencing (RNA-seq) analysis, PAO1 and PaΔH_2_S were cultured in PB medium at 37 °C with shaking at 200 rpm to late log phase. Cells were centrifuged, and the pellets were frozen in liquid nitrogen, and shipped on dry ice to Beijing Novogene Bioinformatics Technology Co., Ltd. (Beijing, China). The subsequent analysis was made by the company. Total RNA was extracted by using a TRIzol^TM^ RNA Purification Kit (12183555, Invitrogen, Waltham, MA USA). Then, 3 μg of total RNA was treated with a Ribo-Zero rRNA Removal Kit (MRZMB 126, Epicentre Biotechnologies). First-strand cDNA was synthesized by using random hexamer primers and M-MuLV Reverse Transcriptase, and second-strand cDNA synthesis was subsequently performed by using DNA polymerase I. RNase H was used to remove RNA. NEBNext index adaptor oligonucleotides were ligated to the cDNA fragments. The cDNA fragments of 150–200 bp in length were purified and amplified via PCR with universal PCR primer and index (X) primer. The library was quantified using an Agilent High Sensitivity DNA assay on an Agilent 267 Bioanalyzer 2100 system and sequenced on the Illumina Hiseq 2500 platform. Trimmed sequence reads were aligned to the *P. aeruginosa* PAO1 genome sequence using Bowtie2-2.2.3. Gene expression was analyzed with the reads per kilobase of coding sequence per million reads (RPKM) algorithm. Differential expression analysis of the two strains was performed using the DESeq R package (1.18.0). Genes with a change fold >2 and a *p*-value <0.05 were considered as significantly differentially expressed.

### 2.8. Reporter Plasmids Construction and Fluorescence Assays

Red fluorescence protein (*mkate*) reporter assays–*E. coli* BL21(DE3) cells were transformed with a plasmid derived from Ptrc99a, containing *lasR,* the promoter of *rhlR*, and *mkate* (Ptrc-*P_lacI_-lasR-P_rhlR_*-*mkate*). Site-directed mutagenesis to convert *lasR* Cys^79^, Cys^188^, Cys^201^, and Cys^203^ to serine was achieved according to a reported method [50]. All primers used in the experiments are listed in Appendix A. Using this reporter strain, LasR activity was quantified. Starting with a 1/1000-fold dilution of an overnight culture, the reporter strain was grown in LB medium supplemented with 20 µM 3O-C_12_-HSL and ampicillin (100 μg/mL) at 37 °C with shaking. When cultures were grown to an OD_600nm_ of 2, 300 µM, polysulfide was added. After incubating at 37 °C for an additional 2 h to produce *mkate*, 0.2 mL of the cells was transferred to a 96-well plate and the *mkate* fluorescence was measured by using a SynergyH1 microplate reader. The excitation wavelength was set at 588 nm and the emission wavelength was set at 633 nm.

### 2.9. Protein Expression and Purification

*E. coli* BL21(DE3) carrying the expression plasmid PET30-LasR was grown in LB to an OD_600nm_ of 0.4–0.6, IPTG was added to 0.1 mM. Growth was continued at 18 °C, overnight. Cells were harvested by centrifugation, washed twice with ice-cold lysis buffer (50 mM NaH_2_PO_4_, 300 mM NaCl and 20 mM imidazole, pH 8.0), and disrupted using a high pressure crusher SPCH-18 (STANSTED). The sample was centrifuged and the supernatant was loaded onto nickel-nitrilotriacetic acid (Ni-NTA) agarose resin (Invitrogen). The target protein was purified according to the supplier’s recommendations. The eluted protein was loaded onto PD-10 column (GE) for buffer exchange to 20 mM sodium phosphate buffer (pH 7.6). The purity of the proteins was checked via SDS-PAGE.

### 2.10. Electrophoretic Mobility Shift Assay (EMSA)

A 300-bp DNA probe containing the *rhlR* promoter sequence was obtained using PCR from the genomic DNA. For quantitative binding assays, different amounts of purified LasR, DNA probe, and binding buffer (10 mM Tris, 50 mM KCl, 5% glycerin, pH 8.0) were mixed and incubated at 30 °C for 30 min. The reaction mixture was then loaded onto a 6% native polyacrylamide gel and electrophoresed at 180 V for 1.5 h. The gel was subsequently stained with SYBR green I and photographed with a FlourChemQ system (Alpha Innotech, San Jose, CA USA). For EMSA experiments, LasR purification was done in an anaerobic glove box (YQX-II, Xinmiao Medical Instruments, Shanghai, China), filled with a gas mixture (85% N_2_, 10% H_2_, and 5% CO_2_). The O_2_ level was maintained at <0.1% via palladium catalysis of H_2_ reaction with O_2_ and monitored using gas a detector (ADKS-4, EDKORS, Changzhou, China). The buffers for LasR purification were all degassed, and 10 mM dithiothreitol (DTT) was added, when necessary, and removed by passing a PD-10 desalting column before HS_n_^−^ treatment.

### 2.11. In Vitro Transcription–Translation Analysis

An S30 T7 High-Yield Protein Expression System (Promega #L1110) was used for in vitro transcription–translation analysis. The reactions contained 20 μL of S30 Premix Plus, 18 μL of T7 S30 extract, 2 μL of *E. coli* RNA polymerase (NEB #M0551), 1 μL of RNase inhibitor, 500 ng of DNA template containing P*_rhlR_*-*mkate,* and 800 ng of LasR; RNase-free water was added to bring the volume to 50 μL. LasR was used as untreated, 160 μM HS_n_^−^-treated, and HS_n_^−^-treated LasR, which was then reduced by 30 mM DTT. After being incubated with vigorous shaking at 37 °C for 2 h, the translated *mkate* was diluted and assayed by using an Synergy H1 microplate reader. The excitation wavelength was set at 588 nm, and the emission wavelength was set at 633 nm. Fluorescence intensities from other groups were divided by that of the untreated LasR to calculate the relative expression levels.

### 2.12. LC-MS/MS Analysis of LasR

Three samples, untreated LasR, HS_n_^−^-treated LasR, and DTT-treated LasR, were prepared for mass spectral analysis. Untreated LasR was used as purified and diluted to 1 μg/μL in the 20 mM phosphate buffer (pH 7.6). For DTT-treated LasR, 1 mL of the purified LasR was reacted with 30 μL of 1 M DTT. For HS_n_^−^-treated LasR, 1 mL of the purified LasR (1 μg/μL) was mixed with 8 μL of 20 mM polysulfide. All the samples were incubated at room temperature for 1 h. Then denaturing buffer (0.5 M Tris-HCl, 2.75 mM EDTA, 6 M guanidine-HCl, pH 8.1) and iodoacetamide (IAM) were added to denaturalize LasR and block free thiols. Samples were digested by trypsin (Promega) for 12 h at 37 °C and were subjected to C18 Zip-Tip (Millipore) purification for desalting before analysis by HPLC- tandem mass spectrometry (LC-MS) using a Prominence nano-LC system (Shimadzu, Nishinokyo, Japan) and LTQ-OrbitrapVelos Pro CID mass spectrometer (Thermo Scientific, Waltham, MA, USA). A linear gradient of solvent A (0.1% formic acid in 2% acetonitrile) and solvent B (0.1% formic acid in 98% acetonitrile) from 0% to 100% of solvent B in 100 min was used for elution. Full-scan MS spectra (from 400 to 1800 *m/z*) were detected with a resolution of 60,000 at 400 *m/z*.

### 2.13. Real-Time Quantitative Reverse Transcription PCR (RT-qPCR)

For RT-qPCR, the cells were collected at a defined incubation time, total RNA was extracted by using a TRIzol^TM^ RNA Purification Kit (12183555, Invitrogen), and cDNA was synthesized by the HiScript^®^ II Reverse Transcriptase (Vazyme, Nanjing, China). RT-qPCR was performed by using a Bestar SybrGreen qPCR Mastermix (DBI Bioscience, Shanghai, China) and LightCycler 480II (Roche, Penzberg, Germany). For calculation of the relative expression levels of tested genes, *rplS* was used as the reference gene.

## 3. Results

### 3.1. H_2_S and Sulfane Sulfur Production by P. aeruginosa PAO1 and Its Mutants

*P. aeruginosa* PAO1 contains two *sqr* genes (Pa*sqr1* and Pa*sqr2*) and one *pdo* (Pa*pdo*) [16]. In LB medium, the wild type did not release H_2_S, but its mutant (Pa3K) with the three H_2_S-metabolic genes (*sqr1*, *sqr2,* and *pdo*) being knocked out released H_2_S, as detected by filter paper strips containing lead acetate in the gas phase (Figure 1A). *P. aeruginosa* PAO1 also contains four genes (*cbs*, *cse*, *mst,* and *cysI*) capable of producing H_2_S. The four genes were deleted from PAO1 and Pa3K to generate Pa∆H_2_S and Pa7K, respectively; both mutants did not release detectable H_2_S into the gas phase (Figure 1A). However, there was trace sulfide detectable by using the mBBr method in the culture supernatants of the mutants (Appendix A). Intracellular sulfane sulfur was detected with SSP4. PaΔH_2_S and Pa7K clearly contained less sulfane sulfur than PAO1 and Pa3K, especially at 24 h of culturing in LB medium (Figure 1B). The mutants grew equally well compared to the wild type in LB medium. The results indicate that CBS, CSE, MST, and CysI are involved in generating H_2_S and intracellular sulfane sulfur, and the produced H_2_S is oxidized by SQR and PDO.

### 3.2. Virulence Factors and Pathogenicity of PAO1 and Its Mutant Strains

The production of two virulence factors, rhamnolipids and pyocyanin, by *P. aeruginosa* PAO1 and its mutants was assayed. The deletion of H_2_S-oxidizing genes did not affect the production of rhamnolipids and pyocyanin, but the deletion of H_2_S-producing genes decreased, obviously, the production of rhamnolipids and pyocyanin. PaΔH_2_S and Pa7K restored rhamnolipid production with added sulfide, implying that H_2_S participates in the regulation of the production of virulence factors in PAO1 (Figure 2A–D). The deletion of H_2_S-producing genes also decreased the pathogenicity, as PaΔH_2_S and Pa7K almost lost the ability to infect lettuce leaves (Figure 2E).

### 3.3. Linking H_2_S/Sulfane Sulfur to LasR

The RNA-seq results showed that >3000 genes were differentially expressed in PAO1 and PaΔH_2_S, indicating that H_2_S has an immense influence on PAO1 gene expression (Appendix A). The genes related to the production of rhamnolipid and pyocyanin were clearly suppressed in PaΔH_2_S, relative to PAO1, consistent with the observed phenotypes. The potential role of LasR in regulating RhlR and PqsR that control the production of rhamnolipids and pyocyanin was noticed, as the related genes were downregulated in PaΔH_2_S (Appendix A). Upregulated genes are summarized in Appendix A and were dominated by transporters and hypothetical proteins. PAO1 *lasR* null mutant (Pa∆*lasR*) grew as well as the wild type in LB (Appendix A), but it decreased infection on the lettuce leaves (Appendix A).

### 3.4. LasR Senses H_2_S through Sulfane Sulfur

The transcription regulator LasR binds to specific DNA sequences, called *lux* boxes [30,32]. We constructed a reporter plasmid Ptrc-*P_lacI_* -*lasR-P_rhlR_*-*mkate*, containing the *lux* box of the *rhlR* upstream region fused to *mkate*, and introduced it into *E. coli* BL21(DE3). The induction required 3O-C_12_-HSL [40], and the addition of NaHS did not enhance the *mkate* expression (Figure 3A). However, the addition of HS_n_^−^ significantly enhanced the production of *mkate* (Figure 3B). Furthermore, when SQR, which converts H_2_S to sulfane sulfur [51], was also cloned into *E. coli* containing the reporter system, the addition of NaHS enhanced the *mkate* expression (Figure 3A). The results indicate that LasR does not directly sense H_2_S, but senses sulfane sulfur.

LasR contains four cysteine residues (Cys^79^, Cys^188^, Cys^201^, and Cys^203^). They were individually mutated into serine in the reporter plasmid Ptrc-*P_lacI_*-*lasR-P_rhlR_*-*mkate*, producing Ptrc-*P_lacI_*-*lasR/C79S-P_rhlR_-mkate*, Ptrc-*P_lacI_*-*lasR/C188S-P_rh_**_lR_-mkate*, Ptrc-*P_lacI_*-*lasR/C201S-P_rhlR_-mkate*, and Ptrc-*P_lacI_*-*lasR/C203S-P_rhlR_-mkate*. C79S had a limited effect, C188S had a large reduction in activity, and C201S and C203S were inactive (Figure 3B).

### 3.5. Characterization of LasR Modification

LasR was purified with 3O-C_12_-HSL because this apoprotein is insoluble [29,52]. An electrophoretic mobility shift assay (EMSA) showed that HS_n_^−^-treated LasR did not affect its binding to target DNA (Figure 4). However, the HS_n_^−^-treated LasR had a 3.4-fold higher expression of the *mkate* gene than DTT-treated or untreated LasR in a coupled transcription and translation assay (Figure 5), in agreement with the whole-cell reporter assay (Figure 3). These results suggest that the HS_n_^−^-modified LasR is more effective when working with RNA polymerase to initiate transcription.

### 3.6. HS_n_^−^ Modifies Cys^188^, Cys^201^, and Cys^203^ of LasR

In DTT-treated LasR, Cys^188^ in Peptide 1a and Cys^201^, and Cys^203^ in Peptide 2a were unmodified (Figure 6, Appendix A and Appendix A). The thiol groups were blocked by iodoacetamide, indicating that Peptide 1a and Peptide 2a were unmodified. A small fraction of the oxidized form (Figure 6 and Appendix A, Peptide 1b) containing Cys^188^-SOH was present in untreated LasR and HS_n_^−^-treated LasR, but the peak area of Cys^188^-SOH was relatively small. According to the area of the mass spectrogram, HS_n_^−^ treatment extensively modified Cys^188^ with 18% Cys^188^-SSH for Peptide 1c (Figure 6 and Appendix A) and 31% Cys^188^-SSSH for Peptide 1d (Figure 6 and Appendix A); 20% Cys^201^ and Cys^203^ formed a pentasulfur link between the two Cys residues (RS-SSS-SR) (Figure 6 and Appendix A). A peptide containing a disulfide bond between Cys^201^ and Cys^203^ was not found in these samples. The observed and calculated masses of corresponding peptides are given in Appendix A. These results show that HS_n_^−^ readily modifies LasR.

### 3.7. The Expression of lasB, rhlR, and lasI Was Affected by Cellular Sulfane Sulfur

We detected the sulfane sulfur content according to growth stages of *P. aeruginosa* PAO1 in LB medium. The sulfane sulfur contents were high in the mid log phase to stationary phase, and decreased in the declination phase (Figure 7). The expression of *lasB*, *rhlR,* and *lasI*, activated by LasR [31], reached the maximum in the stationary phase and decreased sharply in the declination phase. These results indicate that the intracellular level of sulfane sulfur, which is associated with growth phases, regulates LasR activity.

## 4. Discussion

In bacteria, well-documented examples of signaling mediated by H_2_S and sulfane sulfur are usually related to sulfur metabolism [7,14,53]. The demonstration that H_2_S and sulfane sulfur affect quorum sensing in *P. aeruginosa* PAO1 provides direct support for the previous prediction that H_2_S is a signaling message in bacterium [6]. *P. aeruginosa* PAO1 maintains a high level of cellular sulfane sulfur from the mid log phase to stationary phase (Figure 7). One mL of *E. coli* cells at an OD_600nm_ of 1 corresponds to 3.6 µL of cell volume on average, as determined under 22 growth conditions [54]. By using this conversion factor, the maximum cellular sulfane sulfur content in LB grown *E. coli* cells was calculated to be about 111 µM, from previously reported data [17], and that in *P. aeruginosa* PAO1 can be calculated to be to around 250 µM from the data in Figure 7. The relatively low level of sulfane sulfur in *E. coli* explains why the LasR activity was enhanced by adding H_2_S when the *E. coli* cells contained both SQR (sulfide:quinone oxidoreductase) and the reporter system (Figure 3A) or by adding HS_n_^−^ when the *E. coli* cells contained only the reporter system (Figure 3B).

Bacteria can metabolize L-cysteine directly into H_2_S by cysteine desulfhydrases or to produce sulfane sulfur via the concerted actions of L-cysteine aminotransferase and 3-mercaptopyruvate sulfurtransferase [18]. When bacteria possess SQR, they will also oxidize H_2_S to sulfane sulfur. Sulfane sulfur can be reduced to H_2_S by cellular thiols, such as glutathione, or by thioredoxin and glutareduxin [20,55], or be further oxidized by PDO (persulfide dioxygenase) to sulfite, which spontaneously reacts with sulfane sulfur to produce thiosulfate [11,16]. Since the Pa3K strain and the wild type maintained similar amounts of cellular sulfane sulfur (Figure 1B), the concerted action of SQR and PDO for the oxidation of self-produced H_2_S did not increase the cellular sulfane sulfur in the wild type. In the Pa3K mutant without SQR and PDO, the self-produced H_2_S evaporated into the gas phase, as detected by the filter paper containing lead acetate (Figure 1A). *E. coli* without SQR and PDO has been shown to mainly use L-cysteine metabolism to maintain cellular sulfane sulfur [18]. Thus, the maintenance of cellular sulfane sulfur requires an active metabolism, and it is not a surprise that the cells in the declination phase contain low sulfane sulfur (Figure 7).

LasR is a QS master regulator in *P. aeruginosa* PAO1, and it activates the production of several extracellular products that benefit the population as a whole. Since it is a costly process, *P. aeruginosa* PAO1 develops strategies to regulate LasR activity, including 3O-C_12_-HSL [35,56], RpoS [57,58], and QscR [38]. Our results suggest that LasR activity is further controlled by cellular sulfane sulfur. Both in vitro and in vivo data indicate that sulfane sulfur-modified LasR is significantly more active than unmodified LasR in the presence of 3O-C_12_-HSL (Figure 3 and Figure 5). The involvement of sulfane sulfur in regulating LasR-mediated QS makes sense, as it varies during growth, becoming significantly lower in the declination phase (Figure 7), when cells enter survival mode and no long need LasR activity. A model whereby LasR activity is regulated by both 3O-C_12_-HSL and cellular sulfane sulfur is proposed (Figure 8), allowing the QS to coordinate with the growth phases. When LasR binds 3O-C_12_-HSL, it may bind to the promoter to recruit RNA polymerase for transcription. However, sulfane sulfur-modified LasR is more active than unmodified LasR in promoting transcription. As several virulence factors are activated by LasR, adequate levels of cellular sulfane sulfur are likely important for the pathogenicity of *P. aeruginosa* PAO1 (Figure 2 and Figure 8).

Sulfane sulfur has been shown to modify several gene regulators in different forms. FisR of *Cupriavidus pinatubonensis* JMP134 forms a tetrasulfide crosslinking [7], while CstR of *Staphylococcus aureus* generates a mixture of di-, tri-, and tetra-sulfur crosslinked species, respectively [9,53]. MexR of *Pseudomonas aeruginosa* PAO1 mainly forms a disulfide bond, with a small portion of trisulfide [15]. OxyR of *Escherichia coli* responds to sulfane sulfur stress via persulfidation of OxyR at Cys^199^ [14]. The LasR modification by HS_n_^−^ is similar, but different, with persulfidation and trisulfidation of Cys^188^ and a pentasulfur link between Cys^201^ and Cys^203^ (Figure 6).

Sulfane sulfur affects gene repressors and activators in different ways in the reported examples to date. The modified repressors, such as CstR, SqrR, and MexR, no longer bind to the target site for the de-repression of the controlled genes [8,9,15]. The modification of activators often leads to increased transcription of the target genes. When FisR is modified by sulfane sulfur, it activates σ^54^ -dependent transcription of sulfide-oxidizing genes for sulfide removal [7]. When OxyR is modified, it increases the transcription of several genes for the removal of high levels of cellular sulfane sulfur [14]. LasR is also a gene activator [34], and it becomes more active in initiating transcription of controlled genes (Figure 5). LasR responds to both H_2_O_2_ and sulfane sulfur with opposite effects. H_2_O_2_ treatment of LasR produces a disulfide bond between Cys^201^ and Cys^203^, disrupting the LasR binding to its target DNA [40], explaining why LasR is sensitive to oxidative stress [59,60]. On the other hand, sulfane sulfur treatment of LasR results in a pentasulfur link between Cys^201^ and Cys^203^, which does not affect the DNA binding (Figure 4) and makes LasR more active in transcription initiation (Figure 3 and Figure 5). The H_2_S_n_-modified LasR may be more resistant to H_2_O_2_ damage, but the detailed activation mechanism warrants further investigation.

Garlic extracts that contain organosulfur compounds, such as diallyl disulfides, have antimicrobial activities [61]. The inhibitory activities of diallyl disulfides are in the order tetrasulfide > trisulfide > disulphide > monosulfide [62], implying that the long chain compounds with additional sulfane sulfur are more inhibitory. Furthermore, *diallyl* disulfide reduces the pathogenicity and biofilm development of *P. aeruginosa* PAO1 by targeting its QS systems [63]. The inhibitory effect of *diallyl* disulfide may be in part due to its ability to release sulfane sulfur [64]. As the added *diallyl* disulfide is up to 1.28 mg per mL (about 8.8 mM) [63], the higher concentration may lead to increased levels of sulfane sulfur in *P. aeruginosa* PAO1. At high concentrations, sulfane sulfur is toxic to microorganisms [14].

## 5. Conclusions

The level of cellular sulfane sulfur varies along with the growth phases of *P. aeruginosa* PAO1. The relatively high level of sulfane sulfur enhances the LasR activity for QS, and a low level inhibits LasR activity, which is likely further reduced by increased oxidative stress, when *P. aeruginosa* PAO1 enters the declination phase. These findings add a new level of control of LasR activity, beside the autoinducer, H_2_O_2_, and other factors [34,35,36,37,38,39,40,41], representing a fine-tuned activity regulated by various cellular factors. The discovery solidifies a key step in establishing the general signaling role of sulfide and sulfane sulfur in bacteria.

## Figures and Tables

**Figure 1 antioxidants-10-01498-f001:**
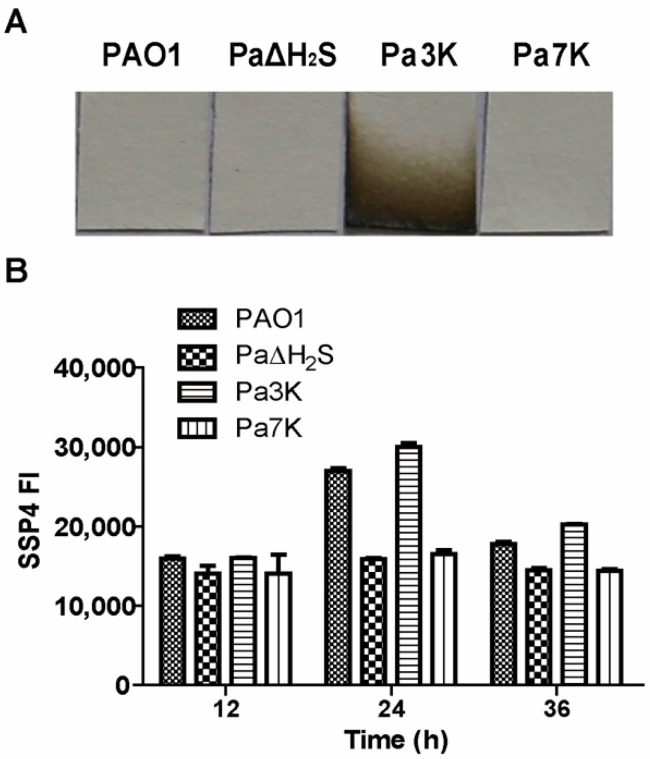
The production of H_2_S and sulfane sulfur in *P. aeruginosa* PAO1 and its mutants. (**A**) Lead-acetate paper strips were used to detect H_2_S in the gas phase during the growth of PAO1 and its mutants in LB medium for 48 h. (**B**) The levels of sulfane sulfur in PAO1 and its mutants were monitored by SSP4 fluorescence.

**Figure 2 antioxidants-10-01498-f002:**
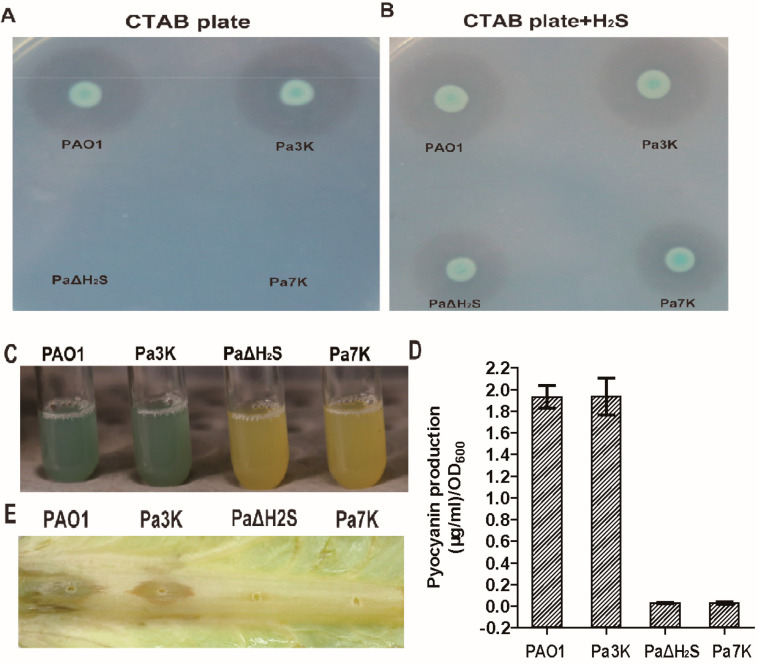
The production of virulence factors by PAO1 and its mutants. (**A**) Rhamnolipid production (clear zone) by PAO1 and its mutants cultured on a CTAB plate. (**B**) Rhamnolipid production by PAO1 and its mutants cultured on a CTAB plate with NaHS being added on the inverted lid. (**C**) Pictures of the cultures of PAO1 and its mutants in PB medium at 37 °C and stationary phase. The green was due to pyocyanin. (**D**) Spectrophotometric quantitation of extracted pyocyanin from the culture supernatants (**C**). Data are averages of three experiments with standard deviations. (**E**) Infection of PAO1 and its mutants on lettuce.

**Figure 3 antioxidants-10-01498-f003:**
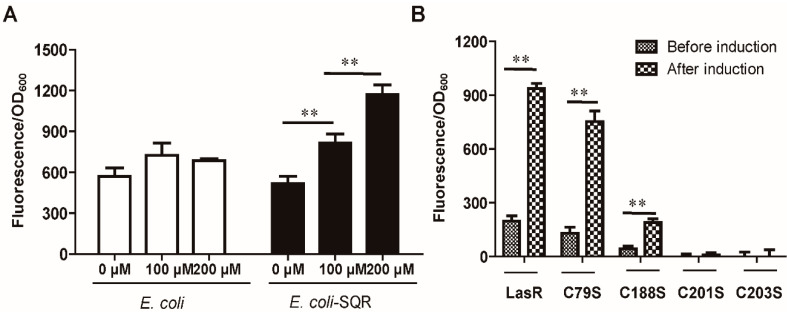
The expression of *mkate* from P*_rhlR_* via LasR activation was significantly enhanced by sulfane sulfur. (**A**) *E. coli* (pTrc-*P_lacI_-lasR-P_rhlR_*-*mkate*)(pBBR1mcs2) or *E. coli* (pTrc-*P_lacI_-lasR-P_rhlR_*-*mkate*)(pBBR1mcs2-SQR) was treated by adding 0, 100, or 200 µM NaHS. Kanamycin and ampicillin were added to maintain the plasmids. (**B**) *E. coli* (pTrc-*P_lacI_-lasR-P_rhlR_*-*mkate*) containing LasR or its mutant with the Cys mutation was treated with 300 µM HS_n_^−^. Ampicillin was added to maintain the plasmid. The *E. coli* cells were culture in LB medium containing 20 μM 3O-C_12_-HSL, while the control contained no 3O-C_12_-HSL. The background fluorescence of the control was subtracted. Data are averages and standard deviations of three experiments. T-tests were performed. ** indicates that the samples were significantly different (*p* < 0.01).

**Figure 4 antioxidants-10-01498-f004:**
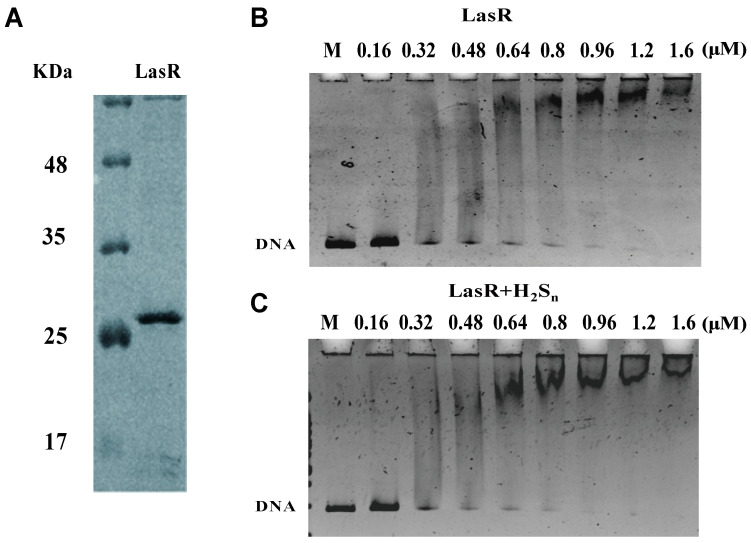
Impact of sulfane sufur on LasR binding to DNA. (**A**) SDS analysis of purified LasR. EMSA analysis of LasR (**B**) and HS_n_^−^-treated LasR (**C**) under anaerobic conditions in the presence of a DNA probe (8 nM) containing the LasR-binding sequence (P*_rhlR_*).

**Figure 5 antioxidants-10-01498-f005:**
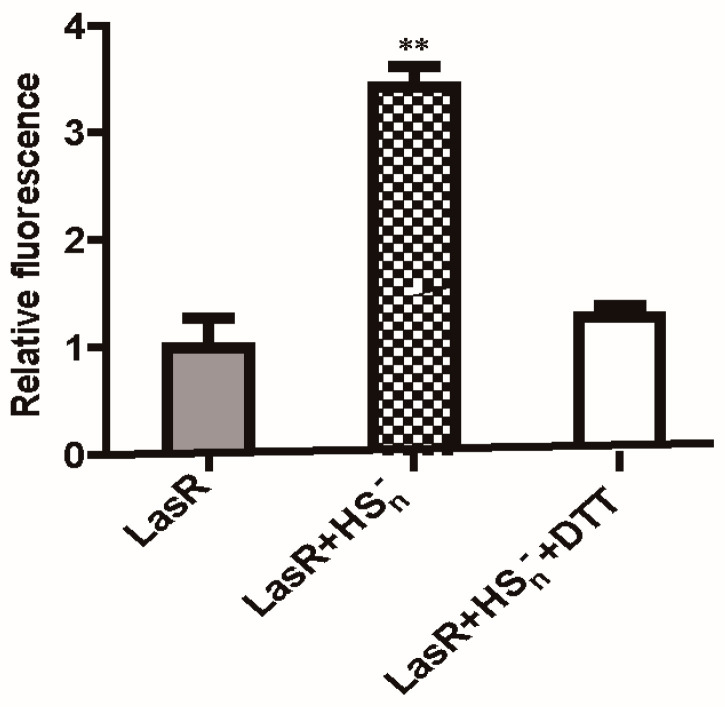
In vitro transcription–translation analysis of LasR activity. Purified LasR, HS_n_^−^-treated LasR, and HS_n_^−^ treated LasR that was then reduced by DTT were used to initiate the transcription and translation from a DNA template containing P*_rhlR_*-*mkate*. The produced fluorescent protein, *mkate,* was detected via its fluorescence. Data are averages of three experiments with standard deviations. T-tests were performed. ** indicates that the sample was significantly different from untreated LasR (*p* < 0.01).

**Figure 6 antioxidants-10-01498-f006:**
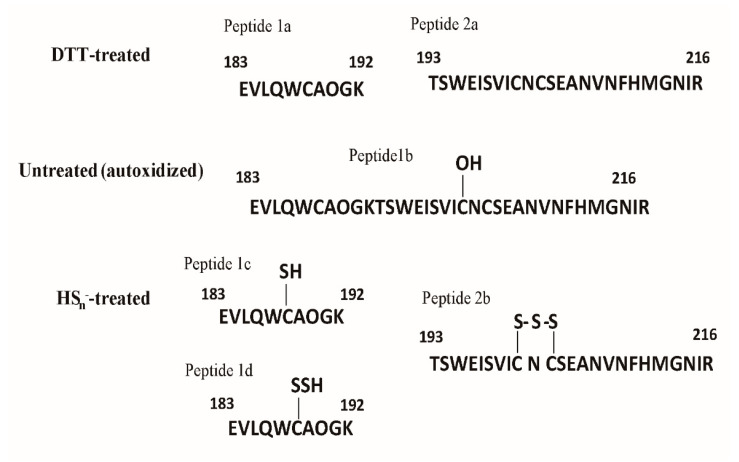
LTQ–Orbitrap tandem mass analysis of HS_n_^−^-reacted LasR. LTQ–Orbitrap tandem mass spectrometry analysis of untreated, DTT-treated, and HS_n_^−^-treated LasR. Peptide 1a and Peptide 2a were found in all samples; Peptide 1b (Cys-SOH) was present in untreated and HS_n_^−^-treated LasR; Peptides 1c (R-SSH), Peptide 1d (R-SSSH), and Peptide 2b (RS-SSS-SR) were detected in HS_n_^−^-treated LasR (Appendix A).

**Figure 7 antioxidants-10-01498-f007:**
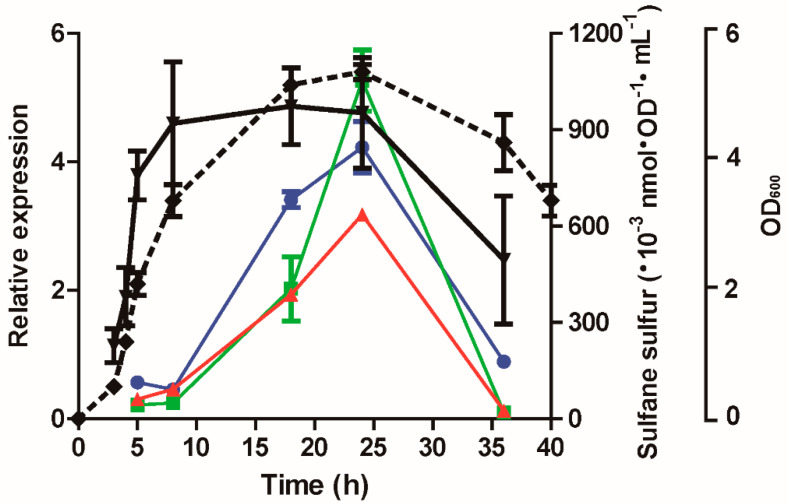
The intracellular levels of sulfane sulfur and the expression of *lasB, rhlR,* and *lasI,* which are activated by LasR, were associated with growth phases. Cell density (♦), intracellular sulfane sulfur contents (▼), and the transcripts of *lasB* (■), *rhlR* (▲), and *lasI* (●) were analyzed at different growth stages (Dash line, OD_600nm_). The *rplS* transcript was used as a reference to calculate the relative expression. Data are averages of three experiments with standard deviations.

**Figure 8 antioxidants-10-01498-f008:**
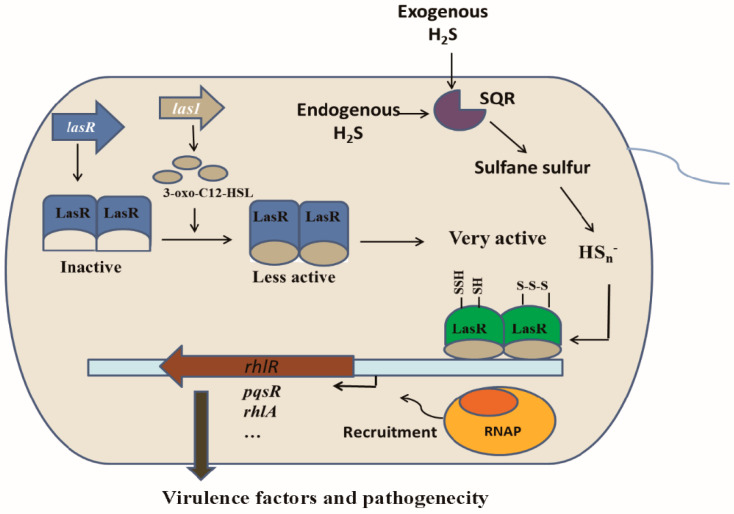
The proposed model of LasR sensing 3O-C_12_-HSL and sulfane sulfur in *P. aeruginosa* PAO1. HS_n_^−^ modifies LasR and enhances its activity.

## Data Availability

Data is contained within the article and Appendix A.

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
