# Peer review of "Sulfane Sulfur Regulates LasR-Mediated Quorum Sensing and Virulence in Pseudomonas aeruginosa PAO1"

_antioxidants, 2021, doi:10.3390/antiox10091498_

Round 1

Reviewer 1 Report

General:

vitro and in vivo in italics in abstract and throughout the MS

the reference style used is not according to the Instruction for authors

Abstract:

…is a common cellular component…

…including virulence factors…

Herein, we report…

Introduction:

L39: elemental sulfur

L68-70: please rephrase and complement this sentence. In addition, please include the following reference:

https://pubmed.ncbi.nlm.nih.gov/33406652/

L72: it may bind to its targets…

L78: please rephrase and complement this sentence. In addition, please include the following reference:

https://jidc.org/index.php/journal/article/view/9920

L95: in the declination phase

Methods:

L100: kanamycin

L106-112: please provide more details

L155: what is PBB medium?

Results:

Figure 1. please highlight the most important results

Figure 6. please improve the quality of this figure

Discussion:

some sections of the discussion are quite confusing due to the many abbreviations. Please re-write these sections for more clarity.

Figure 8. the part regarding virulence factors should be complemented…

Please provide a conclusions section for the article, alternatively from the last part of the discussion.

References:

the reference style used is not according to the Instruction for authors

Author Response

Thank you for the constructive suggestions and comments.  

Reviewer I

General:

vitro and in vivo in italics in abstract and throughout the MS

Response: Done.

the reference style used is not according to the Instruction for authors

Response:  Revised.

Abstract:

…is a common cellular component…

Response: Added “a”.

…including virulence factors…

Response: Changed to “virulence”.

Herein, we report…

Response: Change to “Herein”.

Introduction:

L39: elemental sulfur

Response: Done.

L68-70: please rephrase and complement this sentence. In addition, please include the following reference: https://pubmed.ncbi.nlm.nih.gov/33406652/

Response: Done.

L72: it may bind to its targets…

Response: Done.

L78: please rephrase and complement this sentence. In addition, please include the following reference:

https://jidc.org/index.php/journal/article/view/9920

Response: The sentence was rephrased. A new sentence was added, and the reference was cited (Various efforts including using lavender essential oils have been tried to treat multidrug-resistant P. aeruginosa strains [26]).

L95: in the declination phase

Response: Done.

Methods:

L100: kanamycin

Response: Kanamycin.

L106-112: please provide more details

Response: Done.

L155: what is PBB medium?

Response: Changed to Pseudomonas broth (PB).

Results:

Figure 1. please highlight the most important results

Response: Added the highlights.

Figure 6. please improve the quality of this figure

Response: We simplified the figure and revised the description in the text.

Discussion:

some sections of the discussion are quite confusing due to the many abbreviations. Please re-write these sections for more clarity.

Response: Abbreviations were converted to full names, such as GSH to glutathione, or redefied, such as SQR (sulfide:quinone oxidoreductase) and PDO (persulfide dioxygenase).

Figure 8. the part regarding virulence factors should be complemented…

Response: Add “Virulence factors and pathogenicity” at the bottom of the figure. Mentioned the information in the text.

Please provide a conclusions section for the article, alternatively from the last part of the discussion.

Response: Add a conclusion section.

References:

the reference style used is not according to the Instruction for authors

Response: Done.

Reviewer 2 Report

Review of the paper entitled “Sulfane sulfur regulates LasR-mediated quorum sensing and virulence in Pseudomonas aeruginosa PAO1 by  Guanhua Xuan, Chuanjuan Lü, Huangwei Xu, Kai Li, Huaiwei Liu, Yongzhen Xia and Luying Xun.

      Quorum sensing (QS) phenomenon discovered over 50 years ago, plays a crucial role in bacteria cells. Through the synthesis of small signaling molecules, QS systems regulate the expression of virulence factors, biofilm development, production of secondary metabolites and virulence factors by pathogenic bacteria. It is worth noting that research on the development of methods leading to the inhibition of QS systems called Quorum Quenching (QQ) raises great hopes scientists and doctors, because QQ is seen as an alternative strategy to combat bacterial infections that could replace the use of current antibiotics and minimize the development of the resistance mechanism Pseudomonas aeruginosa cells have two systems QS.  It is a system las encompassing the proteins LasR and LasI and the rhl system complex from RhlR and RhlI proteins. The LasR protein is an activator of the las system. This means that reactive sulfur species (RSS) - hydrogen sulfide and sulfane sulfur compunds are “friends” of the bacterium Pseudomonas aeruginosa. Anyway, many literature data show that the presence of RSS in the cells of many pathogenic bacteria increases their virulence. Many scientists explain the mechanism of this “beneficial” action of RSS for bacteria by the fact that RSS protects bacterial cells from the damaging effects of oxidative stress, a process by which many antibiotics kill bacteria.

     Pseudomonas aeruginosa is a ubiquitous, clinically significant, Gram-negative bacillus and an opportunistic human pathogen. The authors have shown that the modification of the LasR protein by sulfane sulfur in Pseudomonas aeruginosa cells leads to an increase the activity of this protein. In my opinion it is  the most important finding of the authors.

     The authors' research is very interesting. This is a good paper. However, I have a few comments.·     

It is well known that Allium plants, such as garlic, onion, leek, shallot and in particular garlic have long been known to be effective in the therapy of infectious diseases. Medicinal properties of garlic have been attributed to organosulfur compounds. The most important initial sulfur compound occurring in the intact garlic bulbs is alliin (S-allylcysteine sulfoxide). When the garlic is crushed, the enzyme allinase in fresh garlic is activated, under the influence of which alliin is hydrolyzed, producing allicin, pyruvic acid and ammonia. Allicin is easily transformed into oil-soluble polysulfides, mostly diallyl disulfide (DADS), diallyl sulfide (DAS), diallyl trisulfide (DATS), diallyl tertasulfide (DTS) and also into higher polysulfides. DATS and higher polysulfides can be direct sulfane sulfur donors whereas DADS can acquire this ability by tautomerization.

     Li et al. indicated that DADS can inhibit Pseudomonas aeruginosa PAO1 pathogenic factors by inactivating the transcription of key genes from three QS systems (las, rhl, and pqs) [Li, Wen-Ru, et al. Diallyl disulfide from garlic oil inhibits Pseudomonas aeruginosa virulence factors by inactivating key quorum sensing genes. Applied microbiology and biotechnology, 2018, 102.17: 7555-7564].  Nakamoto et al. in their good review paper point out that the antimicrobial activities of organosulfur compounds from garlic depend on the number of sulfur atoms in the molecules and are in the order of diallyl tetrasulfide >diallyl trisulfide>diallyl disulfide>diallyl sulfide. Therefore, DASn containing a higher number of sulfur atom than 5 may have more potent activity against bacterial pathogens [Nakamoto, Masato, et al. Antimicrobial properties of hydrophobic compounds in garlic: Allicin, vinyldithiin, ajoene and diallyl polysulfides. Experimental and therapeutic medicine, 2020, 19.2: 1550-1553]. Casella et al. indicated that DADS showed a good antimicrobial activity against Staphylococcus aureus (inhibition zone 15.9 mm), Pseudomonas aeruginosa (inhibition zone 21.9 mm) and Escherichia coli (inhibition zone 11.4 mm) [Casella, Sergio, et al. The role of diallyl sulfides and dipropyl sulfides in the in vitro antimicrobial activity of the essential oil of garlic, Allium sativum L., and leek, Allium porrum L. Phytotherapy Research, 2013, 27.3: 380-383].

     Many literature data indicate that the mechanism of biological and pharmacological activity of garlic can be explained by the fact that sulfane sulfur, the source of which are organosulfur compounds derived from garlic, can interact with cysteine thiol groups in catalytic, structural or regulatory proteins [Banerjee, R. and Maulik, S.K. (2002) Effect of garlic on cardiovascular disorders: a review. Nutr. J. 19, 1–4; Schafer, G., & H Kaschula, C. (2014). The immunomodulation and anti-inflammatory effects of garlic organosulfur compounds in cancer chemoprevention. Anti-Cancer Agents in Medicinal Chemistry (Formerly Current Medicinal Chemistry-Anti-Cancer Agents), 14(2), 233-240; Iciek, M. et al. (2016). S-sulfhydration as a cellular redox regulation. Bioscience reports, 36(2), e00304; Benavides, Gloria A., et al. "Hydrogen sulfide mediates the vasoactivity of garlic." Proceedings of the National Academy of Sciences 104.46 (2007): 17977-17982; and many, many others].

     So it seems to be a kind of paradox. Covalent modification of proteins by sulfane sulfur derived from organosulfur compounds from garlic is responsible for the antimicrobial effect of garlic, and on the other hand, the authors of the presented study showed that covalent modification of proteins by sulfate sulfur in Pseudomonas aeruginosa  cells increases their activity and virulence.

I would like the authors to discuss this problem.

  • It seems that there is a mistake in the section Detection of H2S and sulfane sulfur.

The Authors describe two methods for the determination of sulfane sulfur in bacterial cells.

The first method is to use a reagent SSP4 „that reacts with sulfane sulfur to become fluorescent was used to detect cellular sulfane sulfur in P. aeruginosa cells (Bibli et al., 2018)”. And then the authors wrote: „Cellular sulfane sulfur in P. aeruginosa PAO1 at different growth stages was determined with a new method that use sulfite to convert sulfane sulfur to thiosulfate for quantification (Ran et al., 2019). ... . The produced thiosulfate was detected using the mBBr method (Kolluru et al., 2013)”. Which method did the authors use to determine the level of sulfane sulfur in bacterial cells? I am asking because in the Results section the authors showed the results obtained only with the method using the SSP4 reagent. It should be explained.

  • The authors use two abbreviations that are not explained, namely: CTAB (page 3) and LB (first time page 4). It should be corrected.

Author Response

Thank you for the constructive suggestions and comments.  

     So it seems to be a kind of paradox. Covalent modification of proteins by sulfane sulfur derived from organosulfur compounds from garlic is responsible for the antimicrobial effect of garlic, and on the other hand, the authors of the presented study showed that covalent modification of proteins by sulfate sulfur in Pseudomonas aeruginosa  cells increases their activity and virulence.

I would like the authors to discuss this problem.

Response: We added a new paragraph to discuss this issue at the bottom of the discussion.

  • It seems that there is a mistake in the section Detection of H2S and sulfane sulfur.

The Authors describe two methods for the determination of sulfane sulfur in bacterial cells.

The first method is to use a reagent SSP4 „that reacts with sulfane sulfur to become fluorescent was used to detect cellular sulfane sulfur in P. aeruginosa cells (Bibli et al., 2018)”. And then the authors wrote: „Cellular sulfane sulfur in P. aeruginosa PAO1 at different growth stages was determined with a new method that use sulfite to convert sulfane sulfur to thiosulfate for quantification (Ran et al., 2019). ... . The produced thiosulfate was detected using the mBBr method (Kolluru et al., 2013)”. Which method did the authors use to determine the level of sulfane sulfur in bacterial cells? I am asking because in the Results section the authors showed the results obtained only with the method using the SSP4 reagent. It should be explained.

Response: SSP4 is used to check the relative levels of cellular sulfane sulfur. We were unable to quantify sulfane sulfur with SSP4. For quantification, cellular sulfane sulfur in P. aeruginosa PAO1 at different growth stages was reacted with sulfite to produce thiosulfate that was then quantified by using a reported method (Ran et al., 2019). The information was added to the method section. 

  • The authors use two abbreviations that are not explained, namely: CTAB (page 3) and LB (first time page 4). It should be corrected.

Response: We change to full names and defined the abbreviations as hexadecyltrimethylammonium bromide(CTAB) and lysogeny broth (LB) medium in the text.